# The Reliability and Validity of a Portable Three-Dimensional Scanning System to Measure Leg Volume

**DOI:** 10.3390/s23229177

**Published:** 2023-11-14

**Authors:** Jack Ashby, Martin Lewis, Caroline Sunderland, Laura A. Barrett, John G. Morris

**Affiliations:** 1Department of Sport Science, Nottingham Trent University, Nottingham NG11 8NS, UK; jack.ashby@ntu.ac.uk (J.A.); caroline.sunderland@ntu.ac.uk (C.S.); 2Qualisys AB, 411 05 Gothenburg, Sweden; martin.lewis@qualisys.com; 3Department of Sport, Exercise and Health Sciences, Loughborough University, Loughborough LE11 3TU, UK; l.a.barrett@lboro.ac.uk

**Keywords:** 3D scanning, water displacement, leg volume, reliability, validity

## Abstract

(1) Background: The study examined the reliability (test–retest, intra- and inter-day) and validity of a portable 3D scanning method when quantifying human leg volume. (2) Methods: Fifteen males volunteered to participate (age, 24.6 ± 2.0 years; stature, 178.9 ± 4.5 cm; body mass, 77.4 ± 6.5 kg; mean ± standard deviation). The volume of the lower and upper legs was examined using a water displacement method (the criterion) and two consecutive 3D scans. Measurements were taken at baseline, 1 h post-baseline (intra-day) and 24 h post-baseline (inter-day). Reliability and validity of the 3D scanning method was assessed using Bland–Altman limits of agreement and Pearson’s product moment correlations. (3) Results: With respect to the test–retest reliability, the 3D scanning method had smaller systematic bias and narrower limits of agreement (±1%, and 3–5%, respectively) compared to the water displacement method (1–2% and 4–7%, respectively), when measuring lower and upper leg volume in humans. The correlation coefficients for all reliability comparisons (test–retest, intra-day, inter-day) would all be regarded as ‘very strong’ (all 0.94 or greater). (4) Conclusions: The study’s results suggest that a 3D scanning method is a reliable and valid method to quantify leg volume.

## 1. Introduction

The measurement of limb volume in clinical practice is used to detect oedemas, lymphedemas, carcinomas and fibrosis [1,2,3]. In sporting contexts, the measurement of limb volume can be used to examine muscle growth and the efficacy of training programmes, but also to detect the seriousness of an injury and to establish the magnitude of exercise induced muscle damage [4]. Clearly, the ability to accurately and reliably measure changes in limb volume can have multiple beneficial applications in clinical and sporting contexts.

In both clinical and sporting contexts, previously published research studies have assessed limb circumference and volume using a wide variety of methodological approaches including measuring tape [5,6], water displacement [7,8], magnetic resonance imaging [9], mathematical modelling [4], bioelectrical impedance spectroscopy, computed tomography and optoelectrical infrared scanning (Perometer) [10,11]. However, many of the existing methods currently used to measure limb circumference and volume may have limitations when utilised in particular clinical and sporting settings. While an anthropometric method using a tape measure may be practical and cost effective, its utility is heavily dependent on the researcher’s ability to competently perform the measurements and to perform the measurements at identical locations. Unfortunately, inter- and intra-observer reliability can be poor, and the method may not provide sufficient accuracy to determine relatively small changes in limb volume, particularly if an aim is to compare limb volume over time [12]. Furthermore, calculating limb volume using a tape measure requires multiple measurements on the limb, which are subsequently incorporated into a truncated cone formula that typically overestimates limb volume because the actual geometry of a limb is not typically well represented by a smooth cone shape [10]. Magnetic resonance imaging and computed tomography do not have many of the methodological problems noted above, and so do provide a valid and reliable solution to the problem of accurately measuring limb volume. However, these methods require expensive equipment, expert staff to perform and interpret the scans, are time consuming, and consequently will not be generally accessible in sporting contexts, or even in many clinical situations [13]. Historically, water displacement volumetry has been the ‘gold standard’ and reference method for evaluating limb volumes [14]. This method involves measurement of the amount of water displaced when a limb is submerged in a container full of water. The water displaced is equal to the volume of the submerged limb (Archimedes’ principle). Although water displacement is considered to be cost effective and reliable, it is not without drawbacks: it may be time consuming as it requires the participant to keep their limb motionless to ensure measurement accuracy and to ensure the correct portion of the limb is submerged and it is not suitable for clinical populations with open wounds such as burns or venous ulcers [15,16,17]. Consequently, it is clear that while currently there are a number of different methods that can be used to measure limb volume, considered against the generic methodological requirements of most clinical and sporting situations (reliability, validity, cost, simple implementation in terms of expertise and time, minimal contamination risk), none is without its limitations.

Three-dimensional (3D) scanning has recently been examined as a potential method to measure limb volume [13,18,19]. The advantages of 3D scanning systems are that they are non-invasive, contactless, it is fast to acquire a 3D scan and they have been reported to be “accurate” [20]. Therefore, if 3D scanning systems are time efficient, reliable and valid, this would make them very attractive within clinical and sporting contexts as a greater number of individuals could be assessed compared to other methods, such as water displacement and tape-measure methods. Seminati and colleagues [13] examined the reliability and validity of a structured light 3D scanning system (Artec EVA Scanner, Artec Group, Luxembourg) when measuring the volume of a residual limb model (which was a mould of an amputee’s leg), compared to a 3D laser scanning system (Romer Scanner, Hexagon, Telford, UK), which was used as the criterion measure. Three observers completed three repeated scans of the residual limb models using the Artec EVA and Romer 3D scanning systems. The results demonstrated that the mean percentage error (validity) for the Artec EVA 3D scanning system was 1.4% (~30 mL difference) compared to the Romer 3D scanning system (criterion). Also, for the Artec EVA 3D scanning system, intra-rater and inter-rater repeatability coefficients were 0.5% and 0.7% (typical error of measurement) for residual limb model volume measurements. Thus, demonstrating that the structured light 3D scanning system (Artec EVA scanner) was a valid and reliable method for measuring residual limb model volume. However, only a few studies have used structured light 3D scanning systems with human participants [21,22], and these studies have focused on facial 3D scanning rather than limb volume measurements. Currently, no research has examined the reliability and validity of structured light 3D scanning for measuring lower body limb volume in healthy individuals. However, before adopting this tool in such contexts, it is necessary to determine its reliability and validity with respect to the measurement of lower body limb volume.

The aims of this study were (1) to examine the reliability (test–retest, intra-day and inter-day) of a structured light 3D scanning system (Artec Leo, Artec Group, Luxembourg) and water displacement method for measuring leg volume; and (2) to examine the measurement validity of a structured light 3D scanning system (Artec Leo) for measuring leg volume compared to a water displacement method.

## 2. Materials and Methods

### 2.1. Participants

Fifteen healthy males (age, 24.6 ± 2.0 years; stature, 178.9 ± 4.5 cm; body mass, 77.5 ± 6.8 kg; mean ± standard deviation) volunteered and provided informed consent to participate in the study. The sample size for the study was based on recommendations of Atkinson and Nevill [23], who reported that a minimum of 20 participants are required to conduct a satisfactory reliability study. In this study, 30 individual legs were used to examine the test–retest reliability, intra- and inter-day reliability and validity, which aligns with the aforementioned recommendation. All participants completed a health screen questionnaire before involvement in the study to ensure they had no medical or other conditions that would have prevented them from taking part. Participants were included in the study if they were between the ages of 18 and 40 years and had no existing lower body injury or medical condition. Participants were instructed to refrained from strenuous exercise in the 48 h prior to an experimental trial, to avoid alcohol consumption 24 h prior to an experimental trial and to avoid caffeine consumption on the day of a trial. The study was approved by a University Ethics Committee (Nottingham Trent University Ethical Committee Application for Human Biological Investigation reference number: 559).

### 2.2. Experimental Design

Participants visited the laboratory on three occasions. During the first visit, participants were familiarised with the water displacement and the 3D scanning procedures. The two subsequent experimental trials were performed on two consecutive days at the same time of day. The first experimental trial comprised of two baseline 3D scans of the participant’s leg volume, followed by two measurements of leg volume using water displacement. One hour following baseline measurements, the procedures were repeated to examine intra-day reliability of each volume measurement method. The following day, leg volume was measured using both methods for a third time to examine inter-day reliability of each method (Figure 1). At the start of both experimental trials, urine osmolality was measured to examine hydration status and body mass was also measured.

### 2.3. Leg Volume Locations

The volumes of lower and upper leg segments were measured in the current study. Using semi-permanent ink, the left and right leg of each participant was divided into foot, lower leg and upper leg segments. Initially, a first mark was made 5 cm above the proximal malleolus (A), which separated the foot from the lower leg. A second mark was made on the most proximal aspect of the patella (B), which separated the lower leg and upper leg. Finally, a third mark was placed 60% proximal from the patella mark (C), which defined the upper leg (Figure 2). Each mark was then extended around the circumference of the leg. Using a mark that covered the circumference of the leg assisted segment volume comparisons between measurement methods. Each mark, and the associated circumference, was made whilst the participant was in a standing position, as participants maintained a standing position during the water displacement and 3D scanning volume measurements. Participants were instructed not to erase the marks to ensure identical locations could be used the following day. Prior to performing the 3D scanning, textured tape (3 × 1 cm) was applied to the anterior aspect of each marked segment in line with the original inked mark. This textured tape assisted the tracking of the 3D scanner to identify the texture and geometry of the leg.

### 2.4. 3D Scanning Procedure

The 3D scans were performed using a handheld Artec Leo 3D scanner (Artec Group, Luxembourg, Luxembourg). Two consecutive scans were performed at each of the study’s timepoints: at baseline, 1 h post-baseline (both Trial 1) and 24 h post the baseline (Trial 2). To perform the 3D scanning, participants stood with their legs shoulder-width apart with their arms crossed over their chest. The participants were asked to refrain from any additional movement during the scan to reduce scan error. During the 3D scanning the anterior of the lower body was captured. The scanner was rotated around the body until the lateral, posterior, and medial aspects of the lower body were imaged to create a 3D scan of the participant’s lower body. When scanning, the 3D scanner was held parallel to the participant at an optimal distance between 0.35 and 1.20 m (manufacturer recommendations). All 3D scans were performed by a trained operator. The completed raw 3D scans were subsequently exported, as “.sproj” files, from the 3D scanner to a compatible computer for scan processing.

### 2.5. 3D Scan Processing

The 3D scanner captured geometry and texture data. The geometry data were used to determine the shape of the lower body, whereas the texture data were used to determine the position of the inked markers on the scans and used for lower and upper leg segment selection. The raw lower body 3D images (scans) were processed using Artec Studio 14 software (Artec Group, Luxembourg). The scans were aligned manually in the reference system x (anterior/posterior), y (medial/lateral) and z (vertical). To process the 3D scans, unwanted data such as the waist, feet and floor were removed using an ‘eraser’ tool. Global registration, outlier removal, smooth fusion and small object filter algorithms were applied within the software. The texture of the 3D scan was then implemented, using the ‘texture’ algorithm to allow identification of the leg segment markers (A, B and C). The ‘eraser’ tool was again used to remove data from the scan which were not required for volume analysis, thus any scan data below the segment A line and above the segment C line was removed (Figure 3). A ‘hole filling’ algorithm was then performed which filled any holes within the 3D scan to provide a closed scan. ‘Hole filling’ removed the texture of the 3D scan so the ‘texture’ algorithm was repeated so that each section of the leg could be identified for volume analysis.

### 2.6. 3D Scan Volume Extraction

The volume measurements were performed using Artec Studio 14 software with the ‘measures’ tool. A coordinate axis square was inserted and aligned with the axis of the 3D lower body scan. The coordinate axis square was then located so that it aligned with the B marker segment line on the leg. The coordinate axis square was used to define the origin of the volume data to be measured. Once in position, the volume above and below the coordinate axis square was measured separately, which corresponded to upper leg volume and lower leg volume (Figure 4). The accurate positioning of the coordinate axis square ensured that the same segments were compared between the 3D scanning and the water displacement methods. To calculate the volume of the 3D scan, the programme calculates the volume of tetrahedrons composed from the vertices of each polygon within the scan and the origin of the coordinate system as the 4th vertex. Calculating the sum of each tetrahedron provided the total volume of the 3D scan. The scan leg volumes were exported in mm^3^ and converted to ml using the following conversion: 1000 mm^3^ = 1 mL. Volume was measured for both left and right legs and the 3D scanning procedure was also performed 1 h and 24 h post the baseline 3D scanning, using identical procedures. As two 3D scans were performed at each time point (baseline, 1 h and 24 h), the means of the two scans were used for subsequent analysis.

### 2.7. Water Displacement Procedure

Two custom-built volumeters were used during the water displacement measurements. A small volumeter (42 × 32 × 25 cm) was used to measure the foot volume and a large volumeter (88 × 48 × 48 cm) was used to measure the volume of the lower and upper leg. The volume of both left and right legs was measured. The volumeters were filled with water until the water spilled out of the overflow spout. The water was left to overflow until no more water spilled from the overflow spout. Prior to immersion of the foot and lower leg and upper leg, water temperature was measured using a mercury thermometer (Fisherbrand, Loughborough, UK). The study researcher sought to ensure the temperature of the water was maintained between 20 °C and 32 °C as this temperature range has been shown to have minimal effects on limb volume [24]. For all trials, the water temperature was 30 ± 1 °C (mean ± standard deviation) for both volumeters. To ensure consistency with the volume measurements made using the 3D scanning method, participants performed the water displacement method in a standing position. For each leg segment (A, B and C), two measurements of volume were performed for both left and right legs of the participants. To measure foot volume, participants slowly immersed their foot into the water until the water surface was level with the inked mark (A). Once the excess water had flowed to less than one drip per second [7], the water collection container was removed and weighed using calibrated weighing scales measuring to two decimal places (kg). The volumes were converted to ml using the following conversion: 1 g = 1 mL. The water in the collection container was poured back into the volumeter for the next immersion. Once foot volume measurements on one leg were completed, the foot volume of the other leg was measured using identical procedures. For lower leg volume, participants immersed their leg into the water until the water surface was level with the inked mark (B). The excess water was subsequently collected and weighed. The identical procedure was then performed for the upper leg mark (C). During the volume measurements, participants were instructed to keep their leg as still as possible to reduce water surface tension caused by movement. The water displacement procedure was also performed 1 h and 24 h post the baseline water displacement, using identical procedures. Although the foot volume was measured using water displacement, this volume was not used for the analysis between methods and was only used for the calculation to measure lower and upper leg volume. To calculate the volume of individual sections of the leg, the prior segment(s) and water collection container weight were deducted. For example, for the upper leg volume, the weight of the lower leg, foot and water container were subtracted from the whole leg weight.

### 2.8. Data Analysis

Two volume measurements were performed using the 3D scanning method and then using the water displacement method for each leg segment at baseline, 1 h post- and 24 h post-baseline. The test–retest reliability of both methods was established by calculating the volume difference between the first and second duplicate measurements for all leg segments at baseline, 1 h post- and 24 h post-baseline. To calculate intra-day reliability, the mean volume of duplicate measurements was calculated at baseline and 1 h post-baseline. To calculate inter-day reliability, the mean volume of duplicate measurements was calculated at baseline and 24 h post-baseline. To calculate the validity of the 3D scanning measurements, mean leg volume was compared to the same value measured using the criterion water displacement method.

The mean leg volume was calculated as follows:mean = (measurement 1 + measurement 2)/2(1)

As volume measurements were performed on both legs of each of the 15 participants, 30 individual legs were used to examine the test–retest reliability, intra- and inter-day reliability and validity.

### 2.9. Statistical Analysis

All data were analysed using the Statistical Package for the Social Sciences (SPSS: Version 26, Chicago, IL, USA), GraphPad Prism (GraphPad Software: Version 9.0.2, San Diego, CA, USA) and Microsoft Excel. To assess the test–retest reliability of the 3D scanning method and the water displacement method, the first and second volume measurements for the lower leg and upper leg at baseline, 1 h post- and 24 h post-baseline were assessed using the Bland–Altman limits of agreement method and the raw and logarithmic transformed systematic bias. A total of 95% limits of agreement were calculated [25]. In addition, Pearson’s product moment correlation coefficient was calculated to examine the relationship between duplicate volume measurements of each method and was interpreted as negligible (0.00–0.10), weak (0.10–0.39), moderate (0.40–0.69), strong (0.70–0.89) and very strong (0.90–1.00) [26]. Intra- and inter-day reliability of the 3D scanning method and the water displacement method was examined using identical methods as those used to establish the test–retest reliability. The mean upper and lower leg volume at baseline were compared to the corresponding volume measured at 1 h post-baseline (intra-day) and 24 h post-baseline (inter-day). The validity of the 3D scanning method to measure lower and upper leg volume was compared to the water displacement (criterion) at baseline, 1 h post-baseline and 24 h post-baseline using the Bland–Altman 95% limits of agreement method, and Pearson’s product moment correlation coefficients. Paired samples t-tests were conducted to examine if any differences in hydration status and body mass existed between trial one and trial two. A significance level of *p* < 0.05 was applied throughout.

## 3. Results

### 3.1. 3D Scanning: Test–Retest Reliability (Lower Leg)

The systematic bias in the lower leg varied from just under 15 mL to just over 35 mL when the first and second 3D scanning volume measurements were compared (Table 1); suggesting that the second 3D scanning measurement consistently overestimated the lower leg volume (by 0–1%). The positive correlation between volume measurements varied from 0.98 to 0.99 (very strong correlations).

### 3.2. Water Displacement: Test–Retest Reliability (Lower Leg)

The systematic bias in the lower leg varied from just under −15 mL to just over −55 mL when the first and second water displacement volume measurements were compared (Table 2); suggesting that the second water displacement method consistently underestimated the lower leg volume (by 1–2%). The positive correlation between volume measurements varied from 0.98 to 0.99 (very strong correlations).

### 3.3. 3D Scanning: Test–Retest Reliability (Upper Leg)

The systematic bias in the upper leg varied from just under −10 mL to just over 50 mL when the first and second 3D scanning volume measurements were compared (Table 3); suggesting that the second 3D scanning measurement both overestimated and underestimated the upper leg volume (by 0–1%). The positive correlation between volume measurements was 0.99 (very strong correlations).

### 3.4. Water Displacement: Test–Retest Reliability (Upper Leg)

The systematic bias in the upper leg varied from just under −35 mL to just over −45 mL when the first and second water displacement volume measurements were compared (Table 4); suggesting that the second water displacement method consistently underestimated the upper leg volume (by 1%). The positive correlation between volume measurements varied from 0.97 to 0.99 (very strong correlations).

### 3.5. 3D Scanning: Intra- and Inter-Day Reliability (Lower Leg)

The systematic bias in the lower leg was just under 55 mL when the baseline and 1 h post-baseline 3D scanning volume measurements were compared (intra-day (Table 5)); suggesting that the 1 h post-baseline 3D scanning measurement overestimated the lower leg volume (by 2%). The positive correlation between volume measurements was 0.96 (very strong correlations). The systematic bias in the lower leg was just under −20 mL when the baseline and 24 h post-baseline 3D scanning volume measurements were compared (inter-day (Table 5)); suggesting that the 24 h post-baseline 3D scanning measurement underestimated the lower leg volume (±1%). The positive correlation between volume measurements was 0.97 (very strong correlations).

### 3.6. Water Displacement: Intra- and Inter-Day Reliability (Lower Leg)

The systematic bias in the lower leg was just under 265 mL when the baseline and 1 h post-baseline water displacement volume measurements were compared (intra-day (Table 6)); suggesting that the 1 h post-baseline water displacement measurement overestimated the lower leg volume (by 9%). The positive correlation between volume measurements was 0.82 (strong correlations). The systematic bias in the lower leg was just under 45 mL when the baseline and 24 h post-baseline water displacement volume measurements were compared (inter-day (Table 6)); suggesting that the 24 h post-baseline water displacement measurement overestimated the lower leg volume (by 2%). The positive correlation between volume measurements was 0.87 (strong correlations).

### 3.7. 3D Scanning: Intra- and Inter-Day Reliability (Upper Leg)

The systematic bias in the upper leg was just under 25 mL when the baseline and 1 h post-baseline 3D scanning volume measurements were compared (intra-day (Table 7)); suggesting that the 1 h post-baseline 3D scanning measurement overestimated the upper leg volume (by 1%). The positive correlation between volume measurements was 0.98 (very strong correlations). The systematic bias in the upper leg was just under 5 mL when the baseline and 24 h post-baseline 3D scanning volume measurements were compared (inter-day (Table 7)); suggesting that the 24 h post-baseline 3D scanning measurement overestimated the upper leg volume (±1%). The positive correlation between volume measurements was 0.98 (very strong correlations).

### 3.8. Water Displacement: Intra- and Inter-Day Reliability (Upper Leg)

The systematic bias in the upper leg was just under 85 mL when the baseline and 1 h post-baseline water displacement volume measurements were compared (intra-day (Table 8)); suggesting that the 1 h post-baseline water displacement measurement overestimated the upper leg volume (by 2%). The positive correlation between volume measurements was 0.96 (very strong correlations). The systematic bias in the upper leg was just under −15 mL when the baseline and 24 h post-baseline water displacement volume measurements were compared (inter-day (Table 8)); suggesting that the 24 h post-baseline water displacement measurement underestimated the upper leg volume (±1%). The positive correlation between volume measurements was 0.94 (very strong correlations).

### 3.9. 3D Scanning versus Water Displacement: Validity (Lower Leg)

The systematic bias in the lower leg varied from just under 95 mL to just over 300 mL when the volumes calculated via the 3D scanning method were compared with those calculated from the water displacement method (Table 9); suggesting that the 3D scanning method consistently overestimated the lower leg volume (by 3–10%). The positive correlation between volume measurement methods varied from 0.71 to 0.92 (strong to very strong correlations).

### 3.10. 3D Scanning versus Water Displacement: Validity (Upper Leg)

The systematic bias in the upper leg varied from just under −45 mL to just over −95 mL when the volumes calculated via the 3D scanning method were compared with those calculated from the water displacement method (Table 10); suggesting that the 3D scanning method consistently underestimated the upper leg volume (by 0–2%). The positive correlation between volume measurement methods varied from 0.95 to 0.97 (very strong correlations).

### 3.11. Hydration Status and Body Mass

Paired samples *t*-tests suggested that hydration status was similar between trials one and two (t(14) = −0.317, *p* = 0.756, trial 1: 668 ± 287 vs. trial 2: 685 ± 251 mOsm/kg). Body mass was also similar between trials one and two (t(14) = 1.469, *p* = 0.164, trial 1: 77.4 ± 6.5 vs. trial 2: 77.1 ± 6.5 kg).

## 4. Discussion

This study sought to examine the test–retest, intra-day, and inter-day reliability of a structured light 3D scanning method (Artec Leo), and also a water displacement method, for measuring leg volume. It also examined the measurement validity of the 3D scanning method for measuring leg volume compared to the water displacement method. The study results indicated that test–retest reliability for the lower leg was better for the 3D scanning method compared to the water displacement method; which was evidenced with smaller systematic bias and narrower limits of agreement for the 3D scanning method (±1%, and 4%, respectively) compared to the water displacement method (1–2%, and 5–7%, respectively). The test–retest reliability for the upper leg was also better for the 3D scanning method compared to the water displacement method, certainly for the limits of agreement (±1%, and 3–5% and 1%, and 4–6%, respectively). This basic pattern of smaller systematic bias and narrower limits of agreement for the 3D scanning method compared to the water displacement method was also evident when intra-day and inter-day reliability of the lower and upper leg volumes were examined. In all the reliability analyses, except for intra- and inter-day measurements made on the lower leg using the water displacement method (*r* = 0.82 and 0.87, respectively), the correlation coefficients for all reliability comparisons (test–retest, intra-day, inter-day) would all be regarded as ‘very strong’ (all 0.94 or greater). With respect to the determination of validity at baseline, and at 1 and 24 h post baseline, when the measurements made using 3D scanning on the upper leg were compared directly with the water displacement method, the systematic bias ranged from 0 to −2% and the limits of agreement from 7 to 9%. The corresponding correlation coefficients ranged from 0.95 to 0.97. This suggests the measurements made using 3D scanning were very close to those made using the ‘gold standard’ water displacement method. The systematic bias was noticeably greater in the lower leg measurements (range 3–10%) and the limits of agreement much wider (15–27%). The corresponding correlations also ranged from 0.71 to 0.92; the potential reasons for this are discussed in detail below. Overall, the results of this study suggest that a portable 3D scanning system is a reliable and valid method with which to quantify human leg volume.

A previous study examining reliability in 13 different types of sports medicine and sports science measurements (i.e., grip strength, leg strength, Wingate maximum power and Fitech step test), found that the mean systematic bias and limits of agreement across the measurements were 3 and 38%, respectively [27]. The corresponding systematic bias and limits of agreement in the current study for duplicate leg volume measurements for the 3D scanning method were 1% and 4% for the lower leg, and ±1% and 4% for the upper leg, respectively. For the water displacement method, the systematic bias and limits of agreement were 1% and 6% for the lower leg, and 1% and 5% for the upper leg, respectively. Therefore, in comparison with typical measurements made in sports medicine and sports science, the test–retest reliability of both the 3D scanning and water displacement methods appear to be very good. This supports the contention that both methods are appropriate tools for measuring leg volume in research where repeated measurements are necessary.

Relatively few studies have investigated the reliability and validity of 3D scanning methods to measure the volume of human legs. The test–retest reliability of the 3D scanning method utilised in the current study was better than that noted by McKinnon and colleagues [28], who found that for arm volume measurements the test–retest reliability systematic bias was 174 mL and 451 mL for their 3D laser scanner and water displacement methods, respectively. In the current study, regardless of the leg segment examined, the mean test–retest reliability systematic bias was 23 mL which is similar to that suggested by Seminati and colleagues [13], who found the test–retest reliability of 14 mL when measuring amputee residual limb models using a structured light 3D scanner (Artec EVA). However, the test–retest reliability, using the 3D scanning method in the current study ranged up to 55 mL, which is poorer than the 14 mL found by Seminati and colleagues [13]. However, given that static residual limb models were used by Seminati and colleagues [13] and non-static humans were used in the current study, the small movement (postural sway) of participants during the scanning procedure may reduce the quality of the 3D scan and increase error and ultimately the volume test–retest reliability. Nonetheless, it appears that the test–retest reliability of the 3D scanning method investigated in the current study has comparable reliability to previous approaches in the published literature, which further supports its use for measuring limb volumes.

Although the 3D scanning method examined in the current study demonstrated excellent test–retest reliability when compared to other measurements typically made in sports medicine and sports science, the key issue is actually whether the reliability is sufficient for the use to which the method will be applied; which in this study was to measure leg volume and ultimately to identify changes in leg volume as a result of clinical dysfunction or sporting activities. In clinical practice, diagnostic criteria thresholds for determining lymphedema were reported by Stout and colleagues [29], who proposed four stages for evaluating early lymphedema based on leg volume change: 0–3% (at risk for lymphedema), 3–5% (pre-clinical lymphedema), 5–8% (mild lymphedema) and >8% (moderate-severe lymphedema). However, most studies have used a >10% volume increase as a diagnostic threshold [30,31,32]. In a sporting context, studies of eccentric biased exercise performed by healthy, untrained, individuals have found that total lower leg volume increased by ~3%, 72 h after eccentric exercise [33,34]. The test–retest reliability results of the current study suggest that both the 3D scanning and water displacement methods may be used to measure lymphedema, in clinical practice, as the measurement systematic bias and limits of agreement were always less than the 10% volume change threshold. As such, if 10% changes were discerned using either method, one could be confident that the change in volume was a genuine leg volume change rather than potentially explainable by measurement error. The test–retest systematic bias for the 3D scanning method ranged from ±1% and the limits of agreement ranged from 3 to 5%. As a result, the 3D scanning method may also be used to establish pre-clinical lymphedema (3–5% volume change). Conversely, both methods may not be appropriate to determine leg volume changes following eccentric exercise as although the systematic bias was below the 3% threshold, the limits of agreement were greater than the ~3% volume change previously reported following eccentric exercise [33,34]. Consequently, a 3% change in limb volume measured using the 3D scanning method in the current study could be genuine, but it could easily also be the result of measurement error.

In order to use the current 3D scanning method to assess limb volume changes after an activity such as eccentric exercise there are two possible solutions. One is to improve the reliability of the 3D scanning method (for example, by ensuring the limits of agreement for various test–retest scenarios is below 3%). Finding ways in which to reduce participant sway might be a way to do this for example. A second alternative would be to accept a less ‘conservative’ set of boundaries. For example, the limits of agreement is essentially based on 2 standard deviations. If one could accept a 68.4% limits of agreement (essentially one based on 1 standard deviation), this would essentially halve the current limits of agreement, and therefore, both methods examined in the current study could be appropriate tools for measuring limb volume changes typically associated with eccentric exercise.

Some studies have investigated intra- and inter-day leg volume changes to establish the magnitude of volume change [8,35,36,37]. Pasley and colleagues [8] measured lower leg volume over five consecutive days using a water displacement method. The results showed that Pearson’s correlation coefficients for volume measurements between days varied between *r* = 0.95 and 0.98; which are similar to the correlation coefficients found in the current study between baseline and 24 h post-baseline (inter-day reliability) which were *r* = 0.87 and 0.94 for the lower and upper leg volume, respectively. Both the 3D scanning and water displacement methods showed that volume differences were smallest between baseline and 24 h post-baseline for the lower and upper leg. Perhaps surprisingly, both methods showed that leg volume increased from baseline to 1 h post-baseline measurements and decreased from 1 h post-baseline to 24 h post-baseline measurements, reducing to near-baseline volumes. These observed changes of leg volume suggest that both the 3D scanning and water displacement methods can detect intra- and inter-day volume changes, but the magnitude of change was greater with the water displacement compared to the 3D scanning method. However, given that both the intra- and inter-day reliability for the 3D scanning method had smaller systematic bias and narrower limits of agreement compared to the water displacement for the lower and upper leg, the 3D scanning may be a better method to examine intra- and inter-day volume changes. Interestingly, the observed (intra- and inter-day) changes in leg volume suggest a time effect was present. A possible mechanism explaining the diurnal leg volume increase demonstrated in this study, from baseline to 1 h post baseline, is suggested to be an increase in the volume of interstitial fluid [35]. Additionally, the prolonged standing required for the volume measurement procedures in this study may have facilitated the increase of interstitial fluid in the legs. Previous research has found diurnal lower leg volume changes, with the volume typically increasing throughout the day [35,38]. Engelberger and colleagues [35] measured lower leg volume of obese and non-obese participants, using an optoelectronic scanner (Perometry), in the morning and afternoon of the same day. The results showed that in both groups, lower leg volume increased during the day, with a mean increase of 59 ± 47 mL in obese participants and 54 ± 24 mL in non-obese participants. These results are similar to the intra-day lower leg volume increase found in the current study (52 ± 106 mL) between baseline and 1 h post-baseline measurements, when measured using the 3D scanning method. Conversely, the corresponding lower leg volume increase was much larger when using the water displacement method (263 ± 314 mL). The limits of agreement were consistently narrower for the 3D scanning compared to the water displacement method for the lower and upper leg volume for intra- and inter-day comparisons. Therefore, given that both the 3D scanning and water displacement methods were measuring identical lower and upper leg segments, the results of the current study suggest that the 3D scanning method is a more reliable one for examining both intra- and inter-day leg volume changes.

Stating that anything is a gold standard method is questionable, nonetheless when measuring limb volume, water-displacement-based methods have been regarded as the optimal measurement approach and therefore, the “gold standard” [16,39]. When comparing the 3D scanning method to the water displacement method in the current study, the results for the upper leg demonstrated that systematic bias (0 to −2%) and limits of agreement (7–9%) were similar to the test–retest reliability noted above. However, it must be noted that the respective systematic bias (3–10%) and limits of agreement (15–27%) for the lower leg were much larger. This apparent difference between the upper and lower leg results are also evidenced by the range of weaker *r* values for the lower leg (*r* = 0.71–0.92) compared to the upper leg (*r* = 95–97). When directly comparing the 3D scanning method to the water displacement method during the validity analysis, the differences noted above may have been caused by the testing protocols utilised when measuring the volume of each leg segment. In the current study, measuring the lower leg volume may have been more demanding for the participants as they were required to suspend their lower leg in the water container and support themselves with the other leg, which may have elicited greater movement of the submerged lower leg, in turn, increasing the disruption of the water tension and increasing the spillage of water which ultimately may have increased the error within each measurement. This notion is supported by the consistent underestimation of lower leg volume for the 3D scanning method compared to the water displacement method. Conversely, to measure the upper leg volume, the whole leg was submerged in the water container and participants were able to gently rest their foot on the bottom of the water container. Therefore, the participants were supported by both legs rather than essentially the one leg used for the lower leg volume measurements. As a result, during the measurement of upper leg volume, less water may have been spilt to attain a measurement, meaning the measurement had less error, and therefore was more representative of the true volume of the leg segment.

There may be several advantages which 3D scanning provides over water displacement in clinical and sporting contexts. Water displacement methods are time consuming to conduct, and in the current study each volume measurement lasted between 10 and 20 min, which is consistent with previous reports [8]. The time constraints of water displacement methods may increase the difficulty of measuring leg volume in both clinical and sporting contexts where multiple individuals may require volume assessment in a short time period. Water displacement methods may not be suitable for some medical applications with individuals with open wounds [15,17]. Furthermore, there is an elevated infection transmission risk with water displacement methods, particularly if multiple individuals undergo the volume assessment using the same volumeter. Therefore, the equipment used for the water displacement method must be effectively sterilised (i.e., the volumeter sterilised and the water changed between participants) to reduce the infection risk [3]. The accuracy of the water displacement method used in the current study was largely dependent on the participant’s ability to keep their leg motionless for up to 20 min as any disturbance to the water tension may add error to the measurement [28]. The latter may have impacted the results in the current study as participants were required to stand on one leg and support their balance with their arms when measuring the leg volume. As a result, this may have been a factor that contributed to the lower test–retest reliability of the water displacement compared to the 3D scanning method. It should also be noted that the participants in the current study were relatively young, healthy individuals for whom standing motionless is likely to be much easier than for many older and clinically impaired individuals. The application of 3D scanning to measure leg volume may address some of the limitations found with the water displacement method. In the current study, each 3D scanning measurement lasted between 2 and 4 min, which was significantly faster than the water displacement method (10–20 min); thus, 3D scanning may be more suitable if multiple volume measurements are required on various individuals both in clinical and sporting contexts. Also, minimal contact is required to identify the volume analysis sections when using the 3D scanning method which may reduce the risk of transmitting infections as the scanning procedure is contactless. Finally, although participant movement during the 3D scan may add errors similar to the water displacement method, the time participants are required to stand motionless is substantially reduced and this may have beneficial impacts on some clinical patients who may have difficulty holding a leg still. Furthermore, in the current study, during the 3D scanning participants were able to stand on both legs whilst the lower body was scanned, the legs were then subsequently separated for analysis. Conversely, for the water displacement measurements, participants’ legs could only be measured individually which relied on participants standing predominantly on one leg, particularly for the lower leg measurements. The application of 3D scanning to measure leg volume seems to address the challenges of the water displacement method. This study demonstrated that the test–retest reliability of the 3D scanning method may not be suitable for measuring volume change following exercise-induced muscle damage (typically ~3% change) as the limits of agreement were greater than 3% for the lower and upper leg. Therefore, future research is required to examine if the 3D scanning method could be more reliable, such as reducing the postural sway when scanning a participant to minimise measurement error. With a 1–2% reduction in the 95% limits of agreement for the test–retest reliability, the 3D scanning method would then be sufficiently reliable to measure leg volume changes following exercise-induced muscle damage. In summary, 3D scanning is a viable alternative method to water displacement for measuring limb volumes, and there may be a number of practical advantages of using 3D scanning rather than water displacement methods in many clinical and sporting contexts.

In the current study the application of the 3D scanning methodology was focused on measuring leg volume. However, there are a number of other purposes to which 3D scanning could be applied in clinical and sporting contexts. In clinical contexts, 3D scanning technology could be used to assist prosthetic socket design in amputees [40], measure scare surface area following burn injuries [41], measure venous ulcers for wound management [42] and examine foot deformities [43]. In sporting contexts, 3D scanning technology could be used to examine body morphology and anthropometry in athletes [44,45] and in the design of optimally fitting custom sports clothing [46]. Clearly, in addition to measuring limb volume, 3D scanning could be usefully applied in a large number of other clinical and sporting contexts.

This study is not without limitations. The authors acknowledge that the water displacement methodology is not the only reference method available for measuring limb volume, and computerised tomography could, for example, have been used, and its use could have strengthened the comparisons made in the current study and perhaps provided greater statistical power. In addition, the study adopted three measurement time points for the water displacement and 3D scanning methods (baseline, 1 h post, and 24 h post the baseline) to examine the reliability of each method for measuring leg volume. However, the study could have included measurements over a series of days (e.g., over a week) which would have provided additional useful results. Also, the authors acknowledge that the statistical outcomes could have been enhanced if more measurements were made with both methods at each individual time point (baseline, 1 h post, and 24 h post the baseline).

## 5. Conclusions

In conclusion, the 3D scanning method provided better test–retest reliability than the water displacement method as the 3D scanning method had smaller systematic bias and limits of agreement (±1–1%, and 3–5%, respectively) compared to the water displacement method (1–2% and 4–7%, respectively), for lower leg and upper leg volume measurements. The intra- and inter-day reliability was also better for the 3D scanning method evidenced with narrower limits of agreement for intra-day reliability (3D scanning: 4–7%, and water displacement: 8–20%) and inter-day reliability (3D scanning: 5–6%, and water displacement: 9–16%). Certainly, in the upper leg, the 3D scanning method was also shown to be a valid method for measuring upper leg volume as the systematic bias and limits of agreement were within 10% of volume measurements made using a criterion water displacement method. The results of this study show that a structured light 3D scanning system (Artec Leo) is a reliable and valid tool for measuring leg volume, certainly in most clinical and sporting contexts.

## Figures and Tables

**Figure 1 sensors-23-09177-f001:**
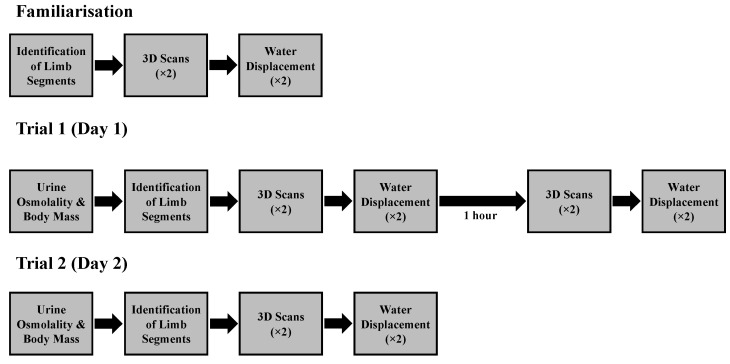
Schematic of the study protocols for the familiarisation trial and the first and second experimental trials.

**Figure 2 sensors-23-09177-f002:**
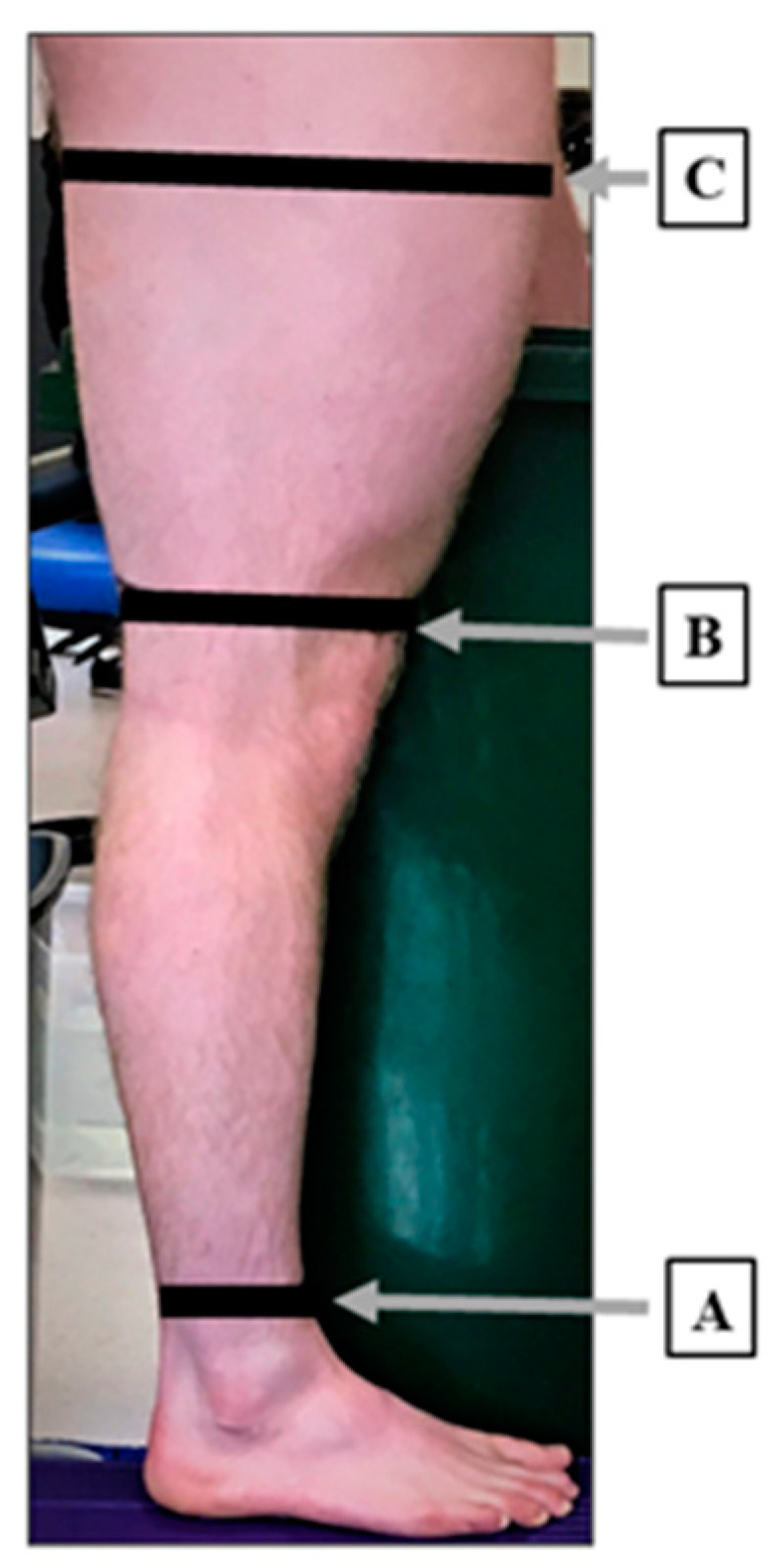
Leg segments A (5 cm above the proximal malleolus), B (the most proximal aspect of the patella) and C (60% proximal from the patella mark) which separated the foot, lower leg and upper leg, respectively.

**Figure 3 sensors-23-09177-f003:**
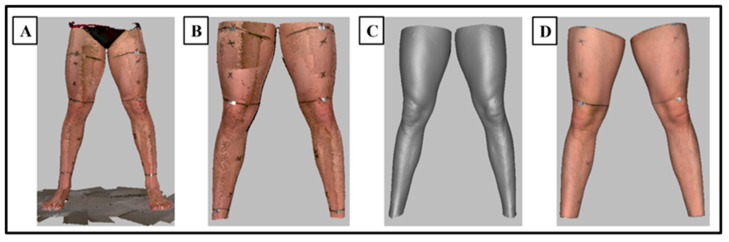
Pictorial representation of the 3D scan processing procedures utilised in the current study: (**A**) the raw unprocessed 3D scan; (**B**) the 3D scan with conservative data removal and applied ‘global registration’ of frames; (**C**) the 3D scan with applied ‘outlier removal’ and ‘smooth fusion’; (**D**) the processed 3D scan with detailed data removal and applied ‘small object filter’, ‘hole filling’ and ‘texture’.

**Figure 4 sensors-23-09177-f004:**
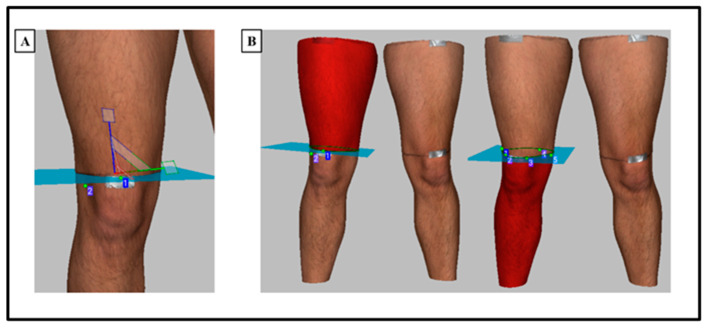
Pictorial representation of the positioning of the coordinate axis square to align with the B marker line (proximal patella) of the leg segment (**A**), and the subsequent volume measurement of the upper and lower leg using the coordinate axis square as the origin for volume measurements (**B**).

**Table 1 sensors-23-09177-t001:** Absolute and relative test–retest reliability of lower leg volume based on two 3D scanning measurements.

	Baseline	1 h Post Baseline	24 h Post Baseline
Sample Size	*n* = 30	*n* = 30	*n* = 30
Measurement 1 (X ± SD) [mL]	3405 ± 330	3466 ± 372	3413 ± 306
Measurement 2 (X ± SD) [mL]	3443 ± 355	3483 ± 369	3427 ± 316
Systematic Bias (mL)	38	17	14
LOA (mL)	134	126	134
Lower; Upper LOA (raw)	−97; 172	−109; 143	−120; 148
Systematic Bias (ln)	1.01	1.01	1.00
LOA (ln)	1.04	1.04	1.04
Lower; Upper LOA (ln)	0.97; 1.05	0.97; 1.04	0.96; 1.05
Pearsons r	0.98 (*p* < 0.001)	0.99 (*p* < 0.001)	0.98 (*p* < 0.001)

X ± SD = mean ± standard deviation; mL = millilitres; ln = logarithmic transformation; LOA = limits of agreement.

**Table 2 sensors-23-09177-t002:** Absolute and relative test–retest reliability of lower leg volume based on two water displacement measurements.

	Baseline	1 h Post Baseline	24 h Post Baseline
Sample Size	*n* = 30	*n* = 30	*n* = 30
Measurement 1 (X ± SD) [mL]	3086 ± 502	3359 ± 519	3149 ± 478
Measurement 2 (X ± SD) [mL]	3052 ± 497	3346 ± 568	3092 ± 478
Systematic Bias (mL)	−33	−13	−56
LOA (mL)	168	187	190
Lower; Upper LOA (raw)	−202; 135	−200; 174	−246; 134
Systematic Bias (ln)	0.99	0.99	0.98
LOA (ln)	1.05	1.06	1.07
Lower; Upper LOA (ln)	0.94; 1.04	0.94; 1.05	0.92; 1.05
Pearsons r	0.99 (*p* < 0.001)	0.99 (*p* < 0.001)	0.98 (*p* < 0.001)

X ± SD = mean ± standard deviation; mL = millilitres; ln = logarithmic transformation; LOA = limits of agreement.

**Table 3 sensors-23-09177-t003:** Absolute and relative test–retest reliability of upper leg volume based on two 3D scanning measurements.

	Baseline	1 h Post Baseline	24 h Post Baseline
Sample Size	*n* = 30	*n* = 30	*n* = 30
Measurement 1 (X ± SD) [mL]	5288 ± 664	5283 ± 685	5280 ± 662
Measurement 2 (X ± SD) [mL]	5343 ± 655	5305 ± 679	5274 ± 671
Systematic Bias (mL)	55	22	−6
LOA (mL)	219	173	176
Lower; Upper LOA (raw)	−164; 274	−151; 195	−183; 170
Systematic Bias (ln)	1.01	1.00	1.00
LOA (ln)	1.05	1.03	1.04
Lower; Upper LOA (ln)	0.97; 1.06	0.97; 1.04	0.96; 1.04
Pearsons r	0.99 (*p* < 0.001)	0.99 (*p* < 0.001)	0.99 (*p* < 0.001)

X ± SD = mean ± standard deviation; mL = millilitres; ln = logarithmic transformation; LOA = limits of agreement.

**Table 4 sensors-23-09177-t004:** Absolute and relative test–retest reliability of upper leg volume based on two water displacement measurements.

	Baseline	1 h Post Baseline	24 h Post Baseline
Sample Size	*n* = 30	*n* = 30	*n* = 30
Measurement 1 (X ± SD) [mL]	5355 ± 658	5461 ± 662	5349 ± 665
Measurement 2 (X ± SD) [mL]	5325 ± 692	5413 ± 673	5308 ± 654
Systematic Bias (mL)	−30	−48	−41
LOA (mL)	315	186	272
Lower; Upper LOA (raw)	−344; 285	−233; 138	−313; 231
Systematic Bias (ln)	0.99	0.99	0.99
LOA (ln)	1.06	1.04	1.06
Lower; Upper LOA (ln)	0.94; 1.05	0.96; 1.03	0.94; 1.05
Pearsons r	0.97 (*p* < 0.001)	0.99 (*p* < 0.001)	0.98 (*p* < 0.001)

X ± SD = mean ± standard deviation; mL = millilitres; ln = logarithmic transformation; LOA = limits of agreement.

**Table 5 sensors-23-09177-t005:** Absolute and relative intra- and inter-day reliability of lower leg volume measured via the 3D scanning method.

	Baseline vs. 1 h Post Baseline	Baseline vs. 24 h Post Baseline
Sample Size	*n* = 30	*n* = 30
3D Scan (X ± SD) [mL]	3404 ± 382	3404 ± 382
3D Scan (X ± SD) [mL]	3456 ± 386	3388 ± 347
Systematic Bias (mL)	52	−16
LOA (mL)	208	191
Lower; Upper LOA (raw)	−155; 260	−207; 174
Systematic Bias (ln)	1.02	1.00
LOA (ln)	1.07	1.06
Lower; Upper LOA (ln)	0.95; 1.08	0.99; 1.05
Pearsons r	0.96 (*p* < 0.001)	0.97 (*p* < 0.001)

3D = three-dimensional; X ± SD = mean ± standard deviation; mL = millilitres; ln = logarithmic transformation; LOA = limits of agreement.

**Table 6 sensors-23-09177-t006:** Absolute and relative intra- and inter-day reliability of lower leg volume measured via the water displacement method.

	Baseline vs. 1 h Post Baseline	Baseline vs. 24 h Post Baseline
Sample Size	*n* = 30	*n* = 30
WD (X ± SD) [mL]	3102 ± 507	3102 ± 507
WD (X ± SD) [mL]	3366 ± 532	3143 ± 462
Systematic Bias (mL)	263	40
LOA (mL)	615	485
Lower; Upper LOA (raw)	−351; 878	−445; 525
Systematic Bias (ln)	1.09	1.02
LOA (ln)	1.20	1.16
Lower; Upper LOA (ln)	0.91; 1.30	0.87; 1.18
Pearsons r	0.82 (*p* < 0.001)	0.87 (*p* < 0.001)

WD = water displacement; X ± SD = mean ± standard deviation; mL = millilitres; ln = logarithmic transformation; LOA = limits of agreement.

**Table 7 sensors-23-09177-t007:** Absolute and relative intra- and inter-day reliability of upper leg volume measured via the 3D scanning method.

	Baseline vs. 1 h Post Baseline	Baseline vs. 24 h Post Baseline
Sample Size	*n* = 30	*n* = 30
3D Scan (X ± SD) [mL]	5311 ± 654	5311 ± 654
3D Scan (X ± SD) [mL]	5334 ± 646	5315 ± 622
Systematic Bias (mL)	23	4
LOA (mL)	224	258
Lower; Upper LOA (raw)	−201; 247	−254; 262
Systematic Bias (ln)	1.01	1.00
LOA (ln)	1.04	1.05
Lower; Upper LOA (ln)	0.97; 1.05	0.95; 1.06
Pearsons r	0.98 (*p* < 0.001)	0.98 (*p* < 0.001)

3D = three-dimensional; X ± SD = mean ± standard deviation; mL = millilitres; ln = logarithmic transformation; LOA = limits of agreement.

**Table 8 sensors-23-09177-t008:** Absolute and relative intra- and inter-day reliability of upper leg volume measured via the water displacement method.

	Baseline vs. 1 h Post Baseline	Baseline vs. 24 h Post Baseline
Sample Size	*n* = 30	*n* = 30
WD (X ± SD) [mL]	5351 ± 657	5351 ± 657
WD (X ± SD) [mL]	5431 ± 638	5340 ± 654
Systematic Bias (mL)	80	−10
LOA (mL)	365	448
Lower; Upper LOA (raw)	−285; 445	−458; 438
Systematic Bias (ln)	1.02	1.00
LOA (ln)	1.08	1.09
Lower; Upper LOA (ln)	0.95; 1.09	0.91; 1.09
Pearsons r	0.96 (*p* < 0.001)	0.94 (*p* < 0.001)

WD = water displacement; X ± SD = mean ± standard deviation; mL = millilitres; ln = logarithmic transformation; LOA = limits of agreement.

**Table 9 sensors-23-09177-t009:** Absolute and relative validity of lower leg volume measured via 3D scanning and water displacement.

	Baseline	1 h Post Baseline	24 h Post Baseline
Sample Size	*n* = 30	*n* = 30	*n* = 30
WD Volume (X ± SD) [mL]	3102 ± 507	3366 ± 532	3143 ± 462
3D Scan Volume (X ± SD) [mL]	3404 ± 382	3456 ± 386	3388 ± 347
Systematic Bias (mL)	302	91	245
LOA (mL)	696	464	544
Lower; Upper LOA (raw)	−394; 998	−373; 554	−299; 789
Systematic Bias (ln)	1.10	1.03	1.08
LOA (ln)	1.27	1.15	1.20
Lower; Upper LOA (ln)	0.87; 1.40	0.90; 1.19	0.90; 1.30
Pearsons r	0.71 (*p* < 0.001)	0.92 (*p* < 0.001)	0.80 (*p* < 0.001)

WD = water displacement; 3D = three-dimensional; X ± SD = mean ± standard deviation; mL = millilitres; ln = logarithmic trans-formation; LOA = limits of agreement.

**Table 10 sensors-23-09177-t010:** Absolute and relative validity of the upper leg volume measured via 3D scanning and water displacement.

	Baseline	1 h Post Baseline	24 h Post Baseline
Sample Size	*n* = 30	*n* = 30	*n* = 30
WD Volume (X ± SD) [mL]	5351 ± 657	5431 ± 638	5340 ± 654
3D Scan Volume (X ± SD) [mL]	5311 ± 654	5334 ± 646	5315 ± 622
Systematic Bias (mL)	−40	−96	−25
LOA (mL)	324	365	408
Lower; Upper LOA (raw)	−364; 284	−491; 299	−433; 383
Systematic Bias (ln)	0.99	0.98	1.00
LOA (ln)	1.07	1.08	1.09
Lower; Upper LOA (ln)	0.93; 1.06	0.91; 1.06	0.91; 1.09
Pearsons r	0.97 (*p* < 0.001)	0.95 (*p* < 0.001)	0.95 (*p* < 0.001)

WD = water displacement; 3D = three-dimensional; X ± SD = mean ± standard deviation; mL = millilitres; ln = logarithmic trans-formation; LOA = limits of agreement.

## Data Availability

The data generated to support the findings of this study are available from the corresponding author upon reasonable request.

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
