# Peer review of "The Reliability and Validity of a Portable Three-Dimensional Scanning System to Measure Leg Volume"

_sensors, 2023, doi:10.3390/s23229177_

Round 1

Reviewer 1 Report

Comments and Suggestions for Authors

The authors conducted a study to examine the reliability and validity of a portable 3D scanning method when quantifying human leg volume.

The manuscript is well written, but it can be further improved.

Introduction:

Page 2, line 69. No 3D abbreviation was introduced after the word three-dimensional but later the abbreviation was used in line 70, 71, 72 and more.

Page 2, line 76. Artec EVA Scanner – manufacturer? Country?

Page 2, line 94. Artec Leo – manufacturer? Country?

Materials and Methods:

Page 3, line 100: How the sample size calculation (n=15) was calculated? Any inclusion and exclusion criteria for the participants?

Page 3, line 111. Participants visited the laboratory on three occasions. However, Figure 1 only stated Trial 1 (Day 1) and Trial 2 (Day 2).

Page 3, line 113. The two subsequent experimental trials were performed on two consecutive days at the same time of day. The first experimental trial comprised of baseline 3D scanning of the participant’s leg volume, followed by assessment of leg volume by water displacement. One hour following baseline measurements, the procedures were repeated to examine intra-day reliability of each volume measurement method. No where in the sentences mentioned that the 3D scanning and water displacements were conducted twice as stated in Figure 1.

Page 4, line 162. The completed raw 3D scans were subsequently exported from the 3D scanner to a compatible computer for scan processing. Export in what file format?

Reviewer 2 Report

Comments and Suggestions for Authors

The objectives and the working protocol are scientifically reasonable, although the practical value of replacing the immersion method with 3D scanning is debatable because a 3D scanner is a costly tool that needs to be optimized in its usage.
The assessment of accuracy and precision with only three measurements may be statistically questionable.
The results would have had greater statistical power if a computerized tomography examination had been used as a reference (obviously indicated for medical reasons, not in the research protocol) and a larger number of measurements per case for both methods (immersion and 3D scanning).
I propose to the authors to reformulate the discussion section to present the additional uses that 3D scanning can have beyond volumetric measurements in the context of medical or sports medical pathology.
